# RoRE: Rotary Ray Embedding for Generalised Multi-Modal Scene Understanding

**Ryan Griffiths, Donald G. Dansereau**
Australian Centre for Robotics,
School of Aerospace, Mechanical and Mechatronic Engineering,
The University of Sydney
{r.griffiths,donald.dansereau}@sydney.edu.au

## Abstract

Transformers have emerged as powerful implicit rendering models, capable of performing geometric reasoning and producing photorealistic novel views in a single feedforward pass. A central challenge in these architectures is how to inject camera parameters into the transformer in a way that generalises across diverse sensing conditions. In this work, we present Rotary Ray Embedding (RoRE), an approach that embeds image patches directly as rays, using a learning based rotary positional embedding (RoPE). This ray-based formulation provides a unified and general representation, improving robustness to unconventional camera geometries and sensing modalities. We evaluate our approach on conventional perspective imagery, fisheye cameras, and multi-modal RGB-thermal setups, showing that a single network can flexibly integrate arbitrary numbers of cameras and modalities into a coherent scene representation. Experiments demonstrate improved generalisation and cross-modal consistency compared to existing methods, highlighting the potential for relative ray-based embeddings to build adaptable, plug-and-play vision systems. Code available at: https://roboticimaging.github.io/RoRE

## 1 Introduction

Recent advances in vision transformers have shown their ability to unify geometry and appearance across multiple views, enabling rapid and accurate scene understanding in 3D perception tasks Jin et al. (2025); Wang et al. (2025). A critical design choice in these architectures is how to inject camera information into the model so that it can accurately align visual tokens with the underlying 3D scene structure. While absolute or relative positional encodings have been proposed, it remains an open question how to best represent camera geometry in a way that is both expressive and generalisable.

This challenge is amplified in heterogeneous settings, where inputs may differ in resolution, field of view, and alignment. Multi-modal scenarios further exacerbate the problem: for instance, RGB and thermal cameras exhibit radically different photometric characteristics while still needing to be integrated into a coherent representation. Existing approaches often make strong assumptions about camera intrinsics, sensor type, or rely on manually engineered fusion strategies Lu et al. (2025), which limits flexibility.

In this work, we are interested in building robust 3D vision models that work for a range of camera families. To that end, we introduce RoRE, a ray-based extension of RoPE Su et al. (2024). Instead of embedding patches by index, RoPE parameterises each patch as a ray, directly encoding where in the scene it is looking. To realise this, we extend RoPE in two key ways: (i) the rotation frequencies are learned rather than fixed, and (ii) asymmetric rotations are applied to break the inherent forward-backward symmetry and encourage more uniform attention across the scene. This formulation allows transformers designed for conventional imagery to naturally extend to new camera geometries and sensing modalities.

Unlike recent pose-free architectures such as DuST3R Wang et al. (2024b), VGGT Wang et al. (2025), and MapAnything Keetha et al. (2025), our approach is designed for settings where camera poses are available or can be easily obtained, for instance, in fixed multi-camera rigs commonly

found on vehicles (Geiger et al., 2013; Ettinger et al., 2021). Pose-free methods are valuable in scenarios where calibration is difficult or impossible, but they typically require substantially larger networks and greater computational resources. In contrast, RoRE leverages available pose information to deliver a more efficient and generalisable solution.

We further demonstrate our method's ability to fuse multi-modal information, specifically RGB-thermal inputs. Thermal cues complement visible imagery and enable perception in challenging conditions such as fog, smoke, darkness, or occlusion, supporting applications including asset inspection and search-and-rescue. A unified embedding of RGB and thermal rays also benefits downstream tasks such as classification, anomaly detection, segmentation, and general multi-modal scene understanding. By operating directly on ray-based inputs from heterogeneous sensors, RoRE provides a shared geometric representation without requiring separate architectures for each modality.

We validate RoRE across five datasets, including perspective and fisheye imagery as well as synthetic and captured multi-modal RGB-thermal datasets. Our experiments demonstrate that ray-based rotary embeddings provide improved generalisation across camera geometries and modalities, while enabling a single network to flexibly integrate arbitrary numbers of heterogeneous inputs.

In summary, this paper makes the following contributions: (1) A novel rotary embedding RoRE that unifies absolute and relative encodings in a ray-based formulation. (2) A multi-modal training scheme using modality-specific tokenisers with shared ray-based embeddings, trained via masked cross-modality prediction for robust fusion. (3) Empirical validation across conventional, fisheye, and multi-modal datasets, showing improved generalisation over state-of-the-art baselines. (4) A synthetic indoor multi-modal dataset of 4,000 scenes with ground-truth poses, released with code for data generation and experiments.

This work paves the way for more general network architectures that can be deployed across diverse camera setups, facilitating broader adoption and practical use.

## 2 RELATED WORK

**Positional Embeddings in Transformers.** Transformers rely on positional encodings to inject structure into sequences of tokens, due to being permutation invariant. Early transformers adopted additive absolute positional encoding (APE) (Vaswani et al., 2017; Devlin et al., 2019), which adds a specific bias to a vector based on a position value. Relative positional encoding (RPE) extends this idea by representing pairwise relationships between tokens. One key feature of this is translational invariance, which leads to improved generalisation. This is typically done with either additive bias terms depending on relative position between tokens (Shaw et al., 2018) or rotary positional embedding where the angle of rotation is based on position (Su et al., 2024).

Vision transformers (Dosovitskiy et al., 2021) have the same challenge in 2D where the position of the patches being encoded need to be injected. This has been done in both APE (Oquab et al., 2023) and RPE. However RoPE has become a popular choice of RPE for embedding the patch location (Heo et al., 2024).

**Geometric Vision.** Geometric vision is a line of work with a goal to integrate information from multiple cameras into unified scene understanding, with a range of tasks such as novel view synthesis (Suhail et al., 2022), pose (Shavit et al., 2021) or depth (Yang et al., 2024a;b). Transformer methods such as DuST3R (Wang et al., 2024b) and VGGT (Wang et al., 2025) demonstrate that large transformer models can reconstruct geometry and synthesise views without explicit 3D supervision. The Large View Synthesis Model (LVSM) (Jin et al., 2025) uses pose information for implicit rendering and achieving strong performance with minimal architectural assumptions. Notably, the decoder-only variant of LVSM reduces the architecture to a single stack of self-attention layers. In this work we adopt the LVSM architecture as the basis for our experiments given its simplicity and strong performance.

**Multi-Modal Vision.** Beyond multiple viewpoints, recent research addresses fusing heterogeneous modalities into a shared representation. They target a range of different tasks such as cross-modal super resolution (Arnold et al., 2024) or glass segmentation with RGB-thermal fusion (Huo et al., 2023). Meng & Fukao (2025) fused RGB and thermal information to produce robust depth estimation. Bachmann et al. (2022) proposed MultiMAE which learns a general-purpose transformer

backbone that can process RGB, depth, and other modalities in a unified token space, however it requires confocal images, which is something we address in this work.

Hassan et al. (2024) adapt NeRFs (Mildenhall et al., 2021) to work with RGB and thermal images while Chen et al. (2024) adapts Gaussian splatting (Kerbl et al., 2023) techniques to work for thermal images and Lu et al. (2025) extends it further to fuse thermal and RGB imagery into one scene understanding. These works demonstrate the benefits of complementary information across sensors and motivate extending view-synthesis transformers to multi-modal inputs, where alignment and fusion strategies remain open challenges. To our knowledge, no work exists that perform feedforward multi-modal novel view synthesis, this is a key development of our work.

**Injecting Pose and Modality Information.** A central question in transformer-based implicit rendering is how to inject camera pose and modality information. To faithfully capture camera pose and viewing geometry Sajjadi et al. (2022) embeds this information through additive means to the tokens. Similarly, Jin et al. (2025) employs Plücker ray embeddings to represent viewing geometry using APE. CaPE (Kong et al., 2024) and GTA (Miyato et al., 2024) investigate conditioning attention on relative camera transformations, based on rotational biases. Li et al. (2025) propose PRoPE which encodes entire camera frustums as relative positional embeddings. These works use the fact that relative embedding should generalise better, however these works move away from the ray-based representations, and this hinders generalisation.

Other approaches have opted to rely only on APE, as in MapAnything (Keetha et al., 2025), where they state the RoPE method tends to lead to unnecessary biases. This is something we aim to mitigate in this work.

For embedding additional modalities VRoPE (Liu et al., 2025), Qwen2-VL (Wang et al., 2024a) and other related approaches extend rotary embeddings to handle a temporal dimension. Multi-MAE (Bachmann et al., 2022) embeds multiple modalities into a single latent space. This is done using modality specific patch encoding on inputs and specific modality heads for outputs, we adopt a similar approach in this work.

## 3 ROTARY RAY EMBEDDING

A central goal of this work is to combine the benefits of relative positional encodings with the generality of ray-based representations. Ray embeddings provide a unified way to represent image patches across diverse camera types, while relative encodings capture geometric relationships between views. To achieve both, we build on the RoPE formulation and introduce key modifications: embedding rays directly, learning rotation frequencies, and using asymmetric position values to reduce attention biases. Using this embedding design, we develop a multi-modal geometric transformer that leverages our proposed RoRE, adapting the architecture to handle heterogeneous sensor modalities and integrate diverse camera inputs into a coherent geometric representation.

### 3.1 PRELIMINARIES

Su et al. (2024) introduced RoPE which is designed to perform rotations of a $d$ dimensional vector $\boldsymbol{x}$ as a way to embed relative positional information into a new vector $x_{rotated}$. Mathematically this is:

$$\boldsymbol{x}_{rotated} = f_{RoPE}(\boldsymbol{x}, m), \tag{1}$$

$$f_{RoPE}(\boldsymbol{x}, m) := \boldsymbol{R}_m^n x, \tag{2}$$

where $\boldsymbol{R}_m^n$ is an $n$ dimensional rotation matrix constructed by multiple $2D$ rotation matrices:

$$R_m^n = \text{diag}\left[ R^{2d}(m\theta_1), R^{2d}(m\theta_2), \ ..., \ R^{2d}(m\theta_{n/2}) \right]_{d\times d}, \tag{3}$$

here $R^{2d}(\theta)$ is the $SO(2)$ rotation matrix with angle $\theta$. The $\theta$ is predefined based on the following:

$$\theta_i = 1000^{-2(i-1)/d}, \tag{4}$$

for $i \in [1, 2, ..., d/2]$. This provides decay of the rotation frequencies (Vaswani et al., 2017). For further details and formal explanation see Su et al. (2024).

While this was originally applied to one dimensional positional information for language models, it has also been applied to two dimensional positional embedding of pixel indices (Weinzaepfel et al., 2023; Wang et al., 2024b; Leroy et al., 2024). To do this the vector is split into two $\boldsymbol{x} = [\boldsymbol{x}_{/2}^{(1)}, \boldsymbol{x}_{/2}^{(2)}]$ where one half is rotated according to some pixel index $u$ and the other half is rotated based on some pixel index $v$. Giving:

$$\boldsymbol{x}_{rotated} = [f(\boldsymbol{x}_{/2}^{(1)}, u), f(\boldsymbol{x}_{/2}^{(2)}, v)]. \tag{5}$$

In our case we do not want to use pixel indices instead we want to encode rays.

## 3.2 EMBEDDING RAYS

In this work, we embed a single ray for each image patch. Specifically, we use the ray corresponding to the patch centre, computed as the average of the rays of all pixels within that patch. This ray is then used to embed the position of a given patch. Building from base RoPE, we now have higher dimensional positions, in our case the 6 that are required to represent the Plücker ray. Which has 3 position (or moment in the case of Plücker coordinates) dimensions $t$ and 3 direction dimensions $d$. Meaning a new strategy needs to developed to embed this information. A straightforward extension is to break up the embedding further into 6 parts $\boldsymbol{x} = [\boldsymbol{x}_{/6}^{(1)}, ..., \boldsymbol{x}_{/6}^{(6)}]$, leading to:

$$x_{rotated} = \left[ f\left(\boldsymbol{x}_{/6}^{(1)}, t_x\right), f\left(\boldsymbol{x}_{/6}^{(2)}, t_y\right), f\left(\boldsymbol{x}_{/6}^{(3)}, t_z\right), f\left(\boldsymbol{x}_{/6}^{(4)}, d_x\right), f\left(\boldsymbol{x}_{/6}^{(5)}, d_y\right), f\left(\boldsymbol{x}_{/6}^{(6)}, d_z\right) \right]. \tag{6}$$

We note, the more dimensions being embedded the more fragmented the vector becomes, essentially putting additional constraints on the latent space. The position and direction components differ fundamentally in magnitude and semantic meaning. The magnitude of the frequencies required for translations values is likely to be different to that of the direction vectors.

We propose another approach by replacing the standard handcrafted base frequencies in Eqn. 4 with learned frequencies for each dimension, superimposing their contributions without fragmenting the embedding space. This allows the network to learn how position dimensions interact with different parts of the latent space. Our frequencies $\boldsymbol{\theta}_{p \times d/2}$ have size $p \times \frac{d}{2}$, where $p = 6$ represents the ray dimensions and $d$ is the query/key token dimension.

The final rotation around a given plane in the $d$ dimensional space is the superposition of all the learned $\boldsymbol{\theta}$ values scaled by their respective ray-position values:

$$R_{RoRE}^{2d}(\mathbf{P}, \boldsymbol{\theta}_i) = R^{2d}\left( \sum_p (\mathbf{P}_p \cdot \boldsymbol{\theta}_{i,p}) \right), \tag{7}$$

where $\boldsymbol{P}_p$ is the position value vector $[t_x, t_y, t_z, d_x, d_y, d_z]$ containing the position values for a patch, $\boldsymbol{\theta}_i$ is the learned frequencies across all position dimensions for a given rotation plane $i \in [1, 2, ..., d/2]$. Our RoRE formulation then becomes:

$$\boldsymbol{x}_{rotated} = f_{RoRE}(\boldsymbol{x}, \boldsymbol{P}, \boldsymbol{\theta}), \tag{8}$$

$$f_{RoRE}(\boldsymbol{x}, \boldsymbol{\theta}, \boldsymbol{P}) := \boldsymbol{R}_p^n \boldsymbol{x}, \tag{9}$$

where

$$\boldsymbol{R}_p^n = \text{diag}[R_{RoRE}^{2d}(\mathbf{P}, \boldsymbol{\theta}_1), R_{RoRE}^{2d}(\mathbf{P}, \boldsymbol{\theta}_2), ... R_{RoRE}^{2d}(\mathbf{P}, \boldsymbol{\theta}_{d/2})]\boldsymbol{x}. \tag{10}$$

The learned $\boldsymbol{\theta}$ parameter is randomly initialised using uniform initialisation between 0 to 0.5, it is left to future research to look into alternative initialisation strategies and how this effects performance.

Fig. 1 shows the learned rotation frequencies. It is encouraging that the model discovers a decay structure similar to the handcrafted schedule, despite being trained without any explicit constraints. This behaviour aligns with the established research: representing ray geometry requires a spectrum of frequencies, with higher-frequency components capturing fine-grained variations and lower-frequency components capturing broader spatial trends. The resulting learned decay therefore mirrors the intended multi-scale behaviour of classical positional embedding, providing evidence that the learnt embedding parameters can autonomously recover a meaningful and interpretable frequency structure.

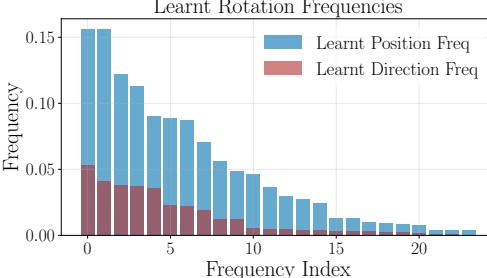 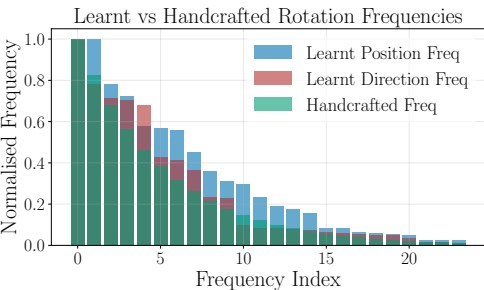

Figure 1: **Comparison of learned vs handcrafted frequency.** Left compares the learned frequency for the position and direction dimension, it shows the magnitude of rotations that has been learned is larger for position than it is for direction. Right is comparing the normalised position and direction frequencies to the standard handcrafted frequency from Eqn. 4. While similar the learned frequencies differ from the handcrafted ones.

There are also clear differences between the position (moment) and direction dimensions, both in frequency decay and scale. This is expected, as the two quantities encode different geometric information. Positional components span a broader normalised range $[0, 1]$, whereas direction vectors vary more subtly due to camera motion constraints and overlapping fields of view. Consequently, the model allocates higher effective frequencies to direction channels and lower ones to position channels. The slightly sharper decay for direction likely reflects the finer rotational relationships between neighbouring rays. Overall, these patterns indicate that RoRE learns a meaningful multi-scale structure without requiring handcrafted schedules.

A key benefit of this method is it removes the manual hyper-parameter selection process that is required for the handcrafted method. We note that the ablation study (Tab. 4) shows very similar performance between the method outlined in Eqn. 6 and the learnt method in Eqn. 8. Due to the benefits of the learned-based approach outlined above we use this method for our proposed approach.

## 3.3 Asymmetric Rotations

Standard RoPE, originally developed for NLP, is designed to emphasise local interactions by causing attention magnitudes to decay as the positional distance between tokens increases Su et al. (2024), which is a natural property of RoPE's formulation. While beneficial for sequence modelling, this behaviour is undesirable in 3D vision, where rays that are far apart in image space may still hold important geometric relationships. To remove this distance-dependent attenuation, we extend an approach taken in VRoPE (Liu et al., 2025) where a shifted negative counterpart of each positional component, ensuring that encoded magnitudes remain consistent across the ray domain. This modification preserves RoPE's rotational properties while preventing the unintended decay in attention, making the embedding better suited to ray-based scene representation. This can be expressed as:

$$\mathbf{P} = \left[\mathbf{t}^+, \mathbf{t}^-, \mathbf{d}^+, \mathbf{d}^-\right] = [\mathbf{t}, -\mathbf{t}, \mathbf{d}, -\mathbf{d}] + [0, b_{shift}, 0, b_{shift}]. \tag{11}$$

where $\mathbf{P}$ is the position vector for a given patch, with $\mathbf{t}$ and $\mathbf{d}$ being the translation and direction components of the ray respectively. The $b_{shift}$ is equal to 1 in our case, as the position $\mathbf{t}$ and direction $\mathbf{d}$ values are normalised to have a maximum value of 1. In practice in the $\boldsymbol{\theta}_{p \times d}$, the $p$ dimension is actually twice the size of the original position vector.

Fig. 2 shows a visual comparison of embedding rays with and without the asymmetric positioning. For this comparison we take unit vectors for query and key tokens and compute the attention score between them after the rotary ray embedding has been applied. This is calculated for 3 images with different poses, with the attention scores being normalised. Without the asymmetric positioning the attention score is not uniform across the patches meaning it is biased toward rays near the query ray. While the asymmetric positions provide a much closer to uniform attention across the frames.

## 3.4 Multi-Modal Geometric Models

A central goal of this work is to design models that are adaptable to diverse sensing modalities while retaining geometric consistency. We achieve this by leveraging transformers' ability to understand

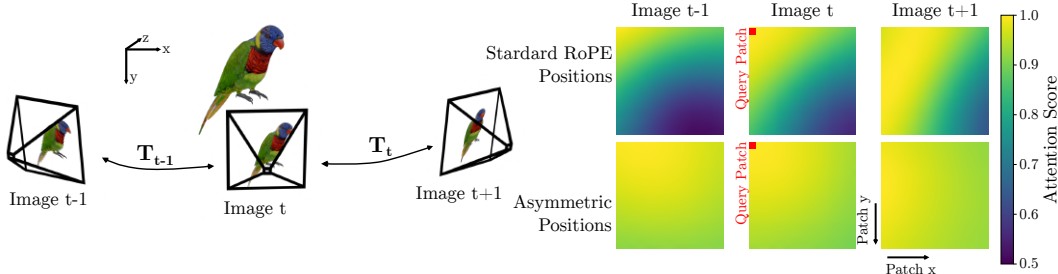

Figure 2: **Attention Comparison.** Attention between frames at different positions using the Plücker parametrisation. The attention score between a query patch, identified in red, and all other patches is shown. Unit query and key vectors are used for this demonstration. The standard RoPE position values bias attention to rays near the query ray. This is problematic because geometric correspondences need not be spatially local. The asymmetric approach removes this bias providing a more uniform attention across possible position values.

multimodal as shown in MultiMAE (Bachmann et al., 2022), with the generic ray based positional embedding. This enables a single forward pass to jointly reason over RGB, thermal, and depth information. Unlike confocal formulations, our method operates directly on posed images, using photometric self-supervision to learn cross-modal correspondences.

Each modality is equipped with its own input tokeniser and output head within an encoder-decoder structure, following the general design of LVSM (Jin et al., 2025). Further architectural details and a diagram are provided in Appendix A.1.2. To represent modality information, we use both absolute embeddings and modality-aware RoRE embeddings. This adds a modality class to the position vector $\boldsymbol{P}$, resulting in $\mathbf{P}^{modality} = \begin{bmatrix} \boldsymbol{P}, C^{modality} \end{bmatrix}$, where $C^{modality}$ is a numerical class modality of the patch. While relative encoding is not directly applicable to discrete modality classes, embedding them within the same framework ensures consistency with the geometric positional encoding.

While our multimodal formulation does not require depth supervision, we include a depth-prediction head to show that RoRE supports explicit geometric estimation. In this work, we use ground-truth depth from our dataset to produce metrically meaningful depth alongside RGB and thermal outputs. A natural extension is to adopt self-supervised depth learning Zhou et al. (2017), enabling operation in settings without ground-truth depth.

Masked input strategies have proven effective in recent vision research, particularly in encouraging models to develop generalisable and semantically rich internal representations (He et al., 2022). Inspired by this, we extend masked input strategies to multi-modal settings, where we mask input image patches based on a fixed masking ratio. This allows the model to learn to interpolate across both missing spatial regions and absent modalities.

Training samples consist of randomly selected context and target views drawn from varying modality combinations to encourage robustness under asymmetric input conditions. The network is trained with a combination of photometric losses (MSE and perceptual) and a depth loss that enforces geometric consistency. While the depth loss is not necessary for RGB-thermal fusion and rendering, it is an additional constraint on geometry and allows for explicit depth estimation from the network. Full details of the training setup and loss formulations are provided in Appendix A.1.2.

## 4 EXPERIMENTS

We perform two sets of experiments: RGB experiments evaluating robustness across camera geometries, and (ii) RGB–thermal experiments evaluating the model's ability to generalisation multi-modal data. Unless otherwise stated, all experiments are run with two input images from a scene with different but known poses. These images correspond to those shown in the qualitative examples.

**Implementation Details.** For the RGB experiments, we adopt the LVSM architecture Jin et al. (2025), which provides a simple yet highly effective baseline for view synthesis. This choice enables controlled and standardised comparisons with alternative methods, though our proposed embedding scheme could be integrated into a wide range of architectures. To reduce computational requirements, we scale the model size down following Li et al. (2025).

Table 1: **Novel view synthesis results.** Results from RealEstate10K (training domain) and DL3DV (unseen but similar domain). All methods perform comparably, with LVSM performing marginally worse and PRoPE performing marginally better. Method marked with † represents concurrent work.

| Method | RealEstate10k | | | DL3DV | | | Iteration Time |
|---|---|---|---|---|---|---|---|
| | PSNR(↑) | SSIM(↑) | LPIPS(↓) | PSNR(↑) | SSIM(↑) | LPIPS(↓) | Seconds |
| LVSM | 26.18 | 0.834 | 0.076 | 19.48 | 0.604 | 0.281 | **1.287** |
| GTA | 26.74 | 0.846 | 0.069 | 19.55 | 0.614 | 0.281 | 1.647 |
| PRoPE† | **26.81** | **0.848** | **0.068** | 19.68 | **0.620** | **0.278** | 1.454 |
| RoRE (ours) | 26.65 | 0.845 | 0.070 | **19.77** | 0.619 | 0.279 | 1.326 |

Table 2: **Quantitative evaluation under varying focal lengths.** Models are tested without retraining by cropping target and query images to simulate changes in camera intrinsics. Methods with stronger representation constraints (GTA and PRoPE) fail to adapt, while LVSM and RoRE remain robust. RoRE consistently outperforms LVSM, demonstrating the advantage of a relative ray-based embeddings for generalisation across intrinsics variations.

| Method | RealEstate10k | | | DL3DV | | |
|---|---|---|---|---|---|---|
| | PSNR(↑) | SSIM(↑) | LPIPS(↓) | PSNR(↑) | SSIM(↑) | LPIPS(↓) |
| LVSM | 21.95 | 0.744 | 0.219 | 19.86 | 0.653 | 0.349 |
| GTA | 14.81 | 0.523 | 0.459 | 14.47 | 0.469 | 0.564 |
| PRoPE | 14.71 | 0.516 | 0.486 | 14.28 | 0.454 | 0.617 |
| RoRE (ours) | **22.66** | **0.770** | **0.211** | **20.31** | **0.678** | **0.335** |

For the multi-modal experiments, we employ a modified architecture in which target and query images interact through cross-attention. This design is motivated by prior work such as DuST3R Wang et al. (2024b) and related models, where cross-attention has proven to be a reliable mechanism for integrating information across views. Additionally as we are learning depth maps we replace the Plücker coordinates with the simpler raymaps, that encode rays as position and direction, instead of moment and direction. Further implementation details, including model configurations, training parameters, hardware setup, and training time, are provided in Appendix A.1.

**Datasets.** We evaluate our approach across a diverse set of datasets spanning conventional perspective imagery, fisheye imagery, and multi-modal RGB-thermal data. For training, we use RealEstate10K (Zhou et al., 2018), a large-scale dataset of posed perspective videos that serves as the primary basis for our models. To assess generalisation to unseen conventional imagery, we further evaluate on DL3DV (Ling et al., 2024), which contains data of a similar type but is not used during training. To test robustness to novel camera geometries, we employ FIORD (Gunes et al., 2025), a fisheye dataset that provides a challenging departure from the perspective imagery seen during training. For multi-modal experiments, we introduce *MultiModalBlender*, a simulated dataset of RGB, thermal and depth images. Large-scale multi-modal datasets are scarce, particularly at the scale required for training transformers, and this dataset enables training with cross-modal fusion. To validate performance qualitatively on real multi-modal imagery, we make use of the Thermal-Gaussian dataset (Lu et al., 2025), which provides data captured in real-world conditions. Together, these datasets allow us to evaluate generalisation across camera geometry, and sensing modality.

**Baselines.** For validating our relative ray-based embedding method we compare to methods of novel view synthesis that perform positional embedding in different ways with different information: LVSM (Jin et al., 2025) which is an absolute positional embedding only method as described in their paper; GTA (Miyato et al., 2024) which uses both ray based absolute embedding and their relative encoding of camera extrinsics; and finally concurrent work PRoPE (Li et al., 2025), which embeds their own camera base absolute embedding and a modified GTA relative embedding that also embed camera intrinsic using a projection matrix. All methods including ours uses the exact same model architecture keeping all parameters the same except for varying the embedding method.

We are unaware of any existing alternative feedforward multi-modal models, as such we do not perform direct comparisons to alternative multimodal feedforward methods for our multi-modal approach instead we show different operating modes to characterise its performance. We evaluate reconstruction quality using PSNR, SSIM, and LPIPS for both RGB and thermal outputs, noting that perceptual metrics such as LPIPS were not originally designed for thermal imagery and therefore cannot be directly compared to their RGB counterparts.

Table 3: **Quantitative evaluation on distorted and fisheye inputs.** Barrel-distorted RealEstate10K images and native FIORD fisheye images are used as inputs without retraining. RoRE generalises robustly to both cases, while GTA and PRoPE fail due to the absence of explicit ray-direction encoding.

| Method | Distorted RE10K | | | Fisheye FIORD | | |
|--------|---------|---------|----------|---------|---------|----------|
| | PSNR($\uparrow$) | SSIM($\uparrow$) | LPIPS($\downarrow$) | PSNR($\uparrow$) | SSIM($\uparrow$) | LPIPS($\downarrow$) |
| LVSM | 21.99 | 0.725 | 0.142 | 22.52 | 0.732 | 0.310 |
| GTA | 18.58 | 0.605 | 0.188 | 11.64 | 0.456 | 0.596 |
| PRoPE | 18.57 | 0.605 | 0.188 | 11.90 | 0.408 | 0.673 |
| RoRE (ours) | **23.96** | **0.802** | **0.124** | **23.55** | **0.746** | **0.284** |

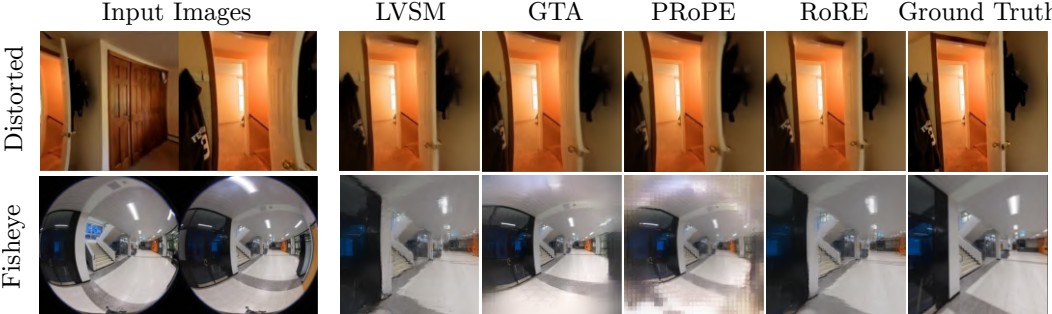

Figure 3: **Qualitative results on distorted and fisheye inputs.** RoRE preserves scene structure under both barrel-distorted perspective images (top) and native fisheye images (bottom), whereas competing methods produce severe artefacts or fail to reconstruct meaningful views.

## 4.1 RoRE Embedding

Tab. 1 reports novel view synthesis results on RealEstate10K and DL3DV. All models are trained on RealEstate10K, with evaluation on the same dataset reflecting in-domain performance, and DL3DV providing an unseen but closely related test set. Across both datasets, the methods achieve broadly comparable results: PRoPE performs slightly better on the training domain, while LVSM is marginally lower. While these results do not highlight a clear advantage for our method, they establish that RoRE remains competitive on standard benchmarks, with its benefits becoming more evident in settings that require greater generalisation, as demonstrated in subsequent experiments.

**Varying Intrinsics.** We evaluate robustness to changes in camera intrinsics by varying the focal length of target and query images through cropping, with randomly chosen magnification of up to 3. This experiment was conducted without retraining the models. As shown in Tab. 2 and Fig. 7 in Appendix A.2.1, methods with stronger representation constraints, such as GTA and PRoPE, fail to adapt. In contrast, LVSM and our RoRE handle these variations effectively, with RoRE consistently outperforming LVSM. While PRoPE can address this case with additional training, our results highlight the inherent advantage of ray-based embeddings, which are naturally invariant to changes in intrinsics.

**Distorted and Fisheye Inputs.**

We next test robustness to non-perspective inputs, using (i) perspective images from RealEstate10K with added barrel distortion and (ii) native fisheye images from FIORD as shown in Tab. 3 and Fig. 3. Distorted inputs are paired with perspective queries, and all evaluations are conducted without retraining. Our method demonstrates consistently stronger generalisation than competing approaches, outperforming LVSM by over 1 dB in PSNR, while GTA and PRoPE fail due to the absence of explicit ray-direction encoding. The fisheye case is particularly relevant, since rectification would reduce field of view; RoRE can handle these inputs directly, capturing the global distortion, though some local inaccuracies remain, see Sec. A.2.2 for further details.

**Discussion.** These results reflect the differing representational biases of the methods. PRoPE uses a constrained, camera-specific formulation based on projection-matrix relative encodings, which aligns well with conventional perspective data such as RE10K, with GTA imposing an even more

Table 4: **Ablation study on RE10K.** The study shows adding the relative embedding in any form improves the performance. The asymmetric positioning provides another modest improvement to performance. Applying the learnt frequency vs handcrafted produces identical results but as stated provides a more general approach. RoRE performs similarly whether the additional absolute embedding is applied or not.

| Absolute Emb. | Relative Emb. | Learnt Frequencies | Asymmetric | PSNR($\uparrow$) | SSIM($\uparrow$) | LPIPS($\downarrow$) |
|:---:|:---:|:---:|:---:|:---:|:---:|:---:|
| ✓ | ✗ | ✗ | ✗ | 26.18 | 0.834 | 0.076 |
| ✓ | ✓ | ✗ | ✗ | 26.56 | 0.843 | 0.071 |
| ✓ | ✓ | ✗ | ✓ | **26.65** | **0.845** | **0.070** |
| ✓ | ✓ | ✓ | ✗ | 26.57 | 0.842 | 0.072 |
| ✗ | ✓ | ✓ | ✓ | **26.65** | 0.843 | **0.070** |
| ✓ | ✓ | ✓ | ✓ | **26.65** | **0.845** | **0.070** |

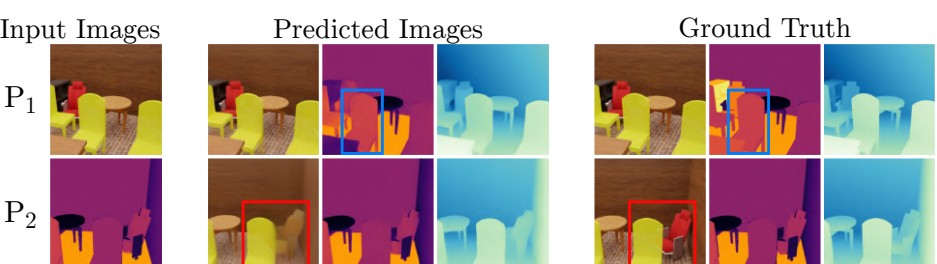

Figure 4: **Qualitative results on the MultiModalBlender dataset.** $P_1$ and $P_2$ denote different camera poses. This figure shows the model's ability to infer missing regions of one modality using cues from another in a geometrically consistent way: (red region) a green chair partially visible in the RGB input is completed using thermal cues, while a red chair absent in RGB inputs has its shape inferred but colour misestimated, (blue region) in the thermal domain, a partially visible chair is reconstructed with accurate geometry and appearance by leveraging RGB information. Depth maps remain consistent across all modality settings.

restricted variant with just the extrinsics. RoRE, by contrast, encodes full rays and learns multi-dimensional frequency interactions, resulting in a more expressive and geometry-agnostic representation. This flexibility leads to substantially stronger generalisation under intrinsics changes, distortion, and fisheye inputs (Table 2, 3). However, its less constrained embedding space can yield slightly lower performance on tightly scoped perspective datasets such as RE10K and DL3DV (Table 1).

**Ablation Study.** We ablate different components of our method, shown in Tab. 4. Firstly, the learnt frequencies refers to the method outlined in Sec. 3.2. The method without it refers to the process outlined in Eqn. 6. Asymmetric refers to the method outlined in Sec. 3.3. Including asymmetric positioning provides a modest increase to performance. Using learned frequencies yields performance comparable to the handcrafted schedule. However, we note that this formulation is a more general solution that removes the need for additional hand tune parameters and handcrafted elements, for this reason our proposed method utilises these learnt frequencies, as they do not impact performance or inference time.

Since the RPE is separate and complementarily to the APE we can use any combination of the embedding methods. The study demonstrates that the performance boost comes from using the RPE compared to the APE, when including both the performance is roughly the same, this is an indication that the network relies on the most appropriate embedding information. Our relative embedding method could work just fine without the absolute embedding, however we do include it in the other experiments as it does show a slightly higher SSIM score.

## 4.2 MULTI-MODAL SCENE UNDERSTANDING

**Simulated Results.** We train a single model capable of handling different modality combinations and numbers of input images. Quantitative results on the MultiModalBlender dataset (Tab. 5) show robust performance across all modality configurations, with slightly reduced depth accuracy for thermal-only inputs due to lower texture information. The lower absolute metrics seen compared to

Table 5: **Quantitative results on the MultiModalBlender dataset.** One model under different input modality configurations (RGB-RGB, RGB-thermal, and thermal-thermal). The results demonstrate that a single model can generalise across all cases.

| Input Images | RGB | | | Thermal | | | Depth | | |
|---|---|---|---|---|---|---|---|---|---|
| | PSNR($\uparrow$) | SSIM($\uparrow$) | LPIPS($\downarrow$) | PSNR($\uparrow$) | SSIM($\uparrow$) | LPIPS($\downarrow$) | AbsRel($\downarrow$) | RMSE($\downarrow$) | $\delta_1(\uparrow$) |
| RGB-RGB | 22.995 | 0.674 | 0.218 | - | - | - | 0.060 | 0.024 | 0.965 |
| RGB-thermal | 21.494 | 0.634 | 0.254 | 20.481 | 0.807 | 0.172 | 0.060 | 0.025 | 0.964 |
| thermal-thermal | - | - | - | 21.662 | 0.825 | 0.153 | 0.065 | 0.027 | 0.959 |

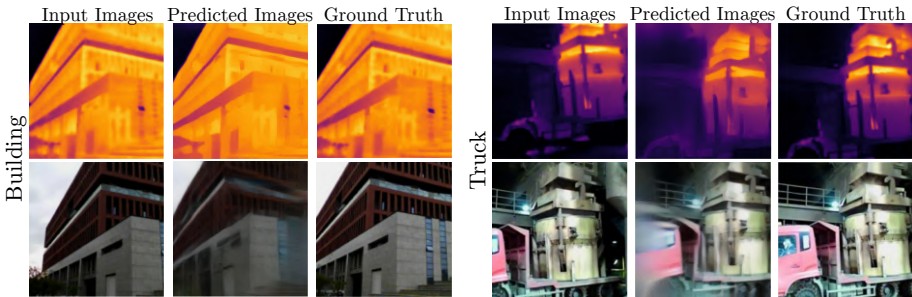

Figure 5: **Qualitative results on the ThermalGaussian dataset** (Lu et al., 2025). The model generates consistent RGB-thermal renderings without additional training.

RealEstate10K are due to more complex motion in the dataset. Qualitative examples (Fig. 4) illustrate the model's ability to fuse modalities: partially visible objects in one modality are completed using cues from the other. For example, (red region) a green chair partially visible in the RGB input is completed using structural cues from the thermal image, while a red chair absent from the RGB inputs has its shape inferred correctly but its colour misestimated. Depth predictions remain coherent, reflecting a unified cross-modal understanding of scene geometry. Additional experiments, including progressive input masking and multi-camera evaluations are provided in Appendix A.2.4.

**Real-world Results.** Fig. 5 shows qualitative renderings on the ThermalGaussian dataset (Lu et al., 2025). Due to the dataset's limited size, we evaluate via inference only. The results demonstrate that the model can process real-world RGB-thermal inputs and produce accurate renderings, even in environments different from the training domain, highlighting its potential for real-world deployment. Some edge effects are present when rendering beyond the spatial extent of inputs; for instance, the truck cabin is extrapolated using nearby visual information. While performance is promising, a simulation-to-real gap remains, we hypothesise, due to differences in scene type, motion, and simulated thermal fidelity, representing an important avenue for future work.

## 5 CONCLUSIONS

We introduced RoRE, a ray-based rotary embedding that generalises effectively across diverse cameras and imaging geometries. Our results show that ray-space embeddings yield stronger performance, rather than extrinsic or projection parameters, under varying camera geometries, such as in the fisheye case. We also extended transformer architectures to support multi-modal inputs and constructed a synthetic RGB-thermal dataset to enable this research. The resulting model achieves unified, geometrically consistent scene understanding from non-confocal images, showing encouraging results on both simulated and real-world data.

One limitation of this formulation is that each patch is represented by a single ray, meaning patch size and orientation are not encoded explicitly. Although these factors are likely inferred implicitly by the network, this does constrain the expressiveness of the representation. Incorporating these parameters directly is an interesting direction for future work. Additional future work includes extending to additional input modalities (e.g., depth or polarisation) and validating multi-modal fusion at larger scales in real-world conditions. This work takes a step toward flexible, plug-and-play vision systems that move beyond conventional camera configurations.

ACKNOWLEDGEMENTS

This research was supported in part through the NVIDIA Academic Grant Program. This research was supported in part by funding from Ford Motor Company.

ETHICS STATEMENT

The development and training of the large-scale models used in this work required substantial computational resources, which in turn consume significant amounts of energy. To promote transparency and awareness around sustainability, we roughly monitored the total energy consumption associated with our development and experiments. See Appendix A.3 for details.

REPRODUCIBILITY STATEMENT

Implementation details have been supplied in Appendix A.1 with configurations, as well as the dataset and code being publicly available.

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

# A APPENDIX

## A.1 IMPLEMENTATION DETAILS

### A.1.1 RGB EXPERIMENTAL DETAILS

**Model Architecture.** The model parameters are outlined in Tab. 6. The total number of parameters for the model is 24.49M.

Table 6: **Multi-modal configuration values.**

| Section | Parameter | Value/Setting |
|---|---|---|
| Model | Layers | 6 |
| | Emdedding Dimension | 768 |
| | Heads | 16 |
| | Positional Embedding | RoRE/LVSM/GTA/PRoPE |
| | Ray Parameterisation | Plücker |
| | Head Type | Linear |
| Patch Embeding | Embedding Type | Linear |
| | Patch Size | 8x8 |
| Dataset | Context Views | 2 |
| | Target Views | 2 |
| Optimizer | Learning Rate | 4.00e-4 |
| | Warm-up Steps | 2500 |
| | Training Steps | 80000 |
| Data Loader | Batch Size (per GPU) | 32 |
| | Image Size | 256x256 |
| Loss | MSE Weight | 1.0 |
| | Perceptual Loss Weight | 0.5 |

**Training.** Training was performed on 2 x RTX6000 Ada GPUs, Training time for a single run was approximately 30 hours.

### A.1.2 MULTI-MODAL EXPERIMENTAL DETAILS

**Model Architecture.** Model parameters are provided in Tab. 7 and a graphical illustration of network architectures is depicted in

Table 7: **Multi-modal configuration values.**

| Section | Parameter | Value/Setting |
|---|---|---|
| Model | Encoder Backbone | ViT Large |
| | Decoder Backbone | ViT Base |
| | Positional Embedding | RoRE |
| | Ray Parameterisation | raymap |
| | Head Type | DPT |
| Patch Embeding | Embedding Type | conv |
| | Patch Size | 16x16 |
| Dataset | Context Views | 8 |
| | Target Views | 6 |
| Optimizer | Learning Rate | 5.00e-5 |
| | Warm-up Steps | 500 |
| Data Loader | Batch Size (train) | 2 |
| | Image Size | 256x256 |
| Loss | MSE Weight ($\lambda_{\mathrm{mse}}$) | 1.0 |
| | LPIPS Weight ($\lambda_{\mathrm{lpips}}$) | 0.05 |
| | Depth Loss Weight ($\lambda_{\mathrm{depth}}$) | 0.75 |
| Masking | Masking Ratio | 50% |

**Training.** We pre-train our network on RealEstate10K, to precondition the network, before training on multi-modal data. This is done because RealEstate10K is a larger dataset and should help generalise. Training was performed on 2 x RTX6000 Ada GPUs, with the initial RealEstate10K

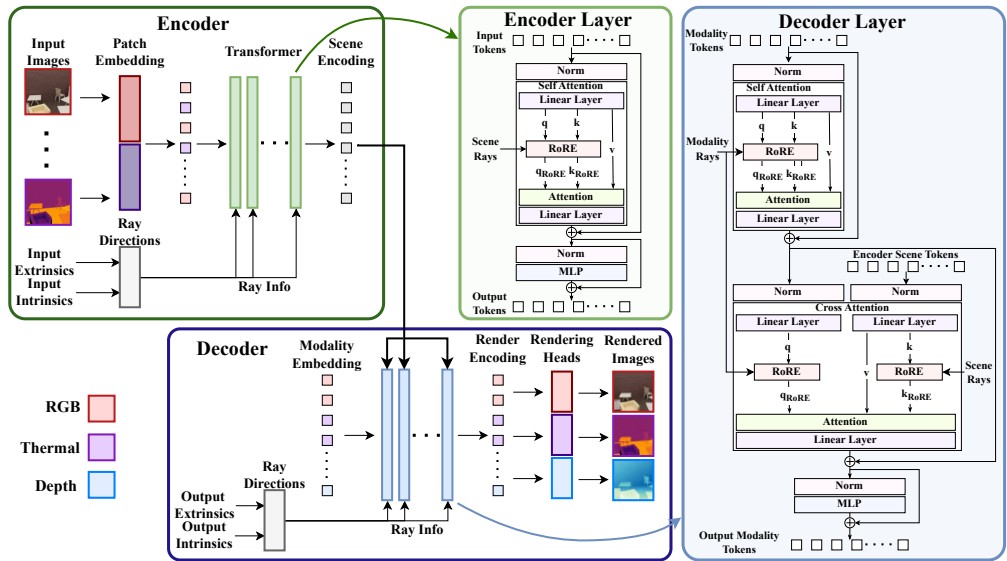

Figure 6: **Multi-modal, multi-camera architecture.** The architecture has two stages, an encoder where images are patchified and encoded into a multi-modal scene encoding. This encoding can then be used by the decoder to render novel views. This architecture makes use of our ray based RoPE to enable the embedding of camera intrinsics and extrinsics. This embedding is used in both the Encoder and Decoder layers. Otherwise these encoder and decoder layers are fairly standard self-attention and cross-attention layers respectively.

pretraining taking around 4 days to complete and the final multi-modal model being trained for 7 days.

**Loss Functions.** To train the network, we employ a combination of photometric and geometric supervision. Specifically, the total loss function consists of two appearance-based losses and one depth-based loss:

$$\mathcal{L} = \lambda_{\text{mse}}\mathcal{L}_{\text{mse}} + \lambda_{\text{lpips}}\mathcal{L}_{\text{lpips}} + \lambda_{\text{depth}}\mathcal{L}_{\text{depth}}. \tag{12}$$

where $\mathcal{L}_{\text{mse}}$ is the MSE loss between predicted and ground-truth RGB or thermal reconstructions, and $\mathcal{L}_{\text{lpips}}$ is the LPIPS, which captures higher-level structural and semantic differences between the reconstructed and reference images. To enforce geometric consistency, we also include a depth loss, $\mathcal{L}_{\text{depth}}$, defined as:

$$\mathcal{L}_{\text{depth}} = \frac{1}{N}\sum_{i=1}^{N}|d_i - \hat{d}_i| + |\nabla d_i - \nabla \hat{d}_i|, \tag{13}$$

where $d_i$ and $\hat{d}_i$ represent the ground-truth and predicted depths at pixel $i$, and $\nabla$ denotes the spatial gradient operator. The loss combines both absolute depth error and gradient-based smoothness, a formulation commonly used in monocular depth estimation encourage accurate relative depth. This loss works well for bounded scene e.g. indoors. If the method was being applied in scenes with large depth values e.g. outside, we would employ losses resilient to the reduced accuracy associated with larger depths.

The scalar weights $\lambda_{\text{mse}}$, $\lambda_{\text{lpips}}$, and $\lambda_{\text{depth}}$ balance the contributions of the respective terms, values for these are given in Tab. 7. This multi-term loss encourages the model to produce reconstructions that are both photometrically accurate and geometrically consistent, which is critical for effective multi-view, multi-modal synthesis.

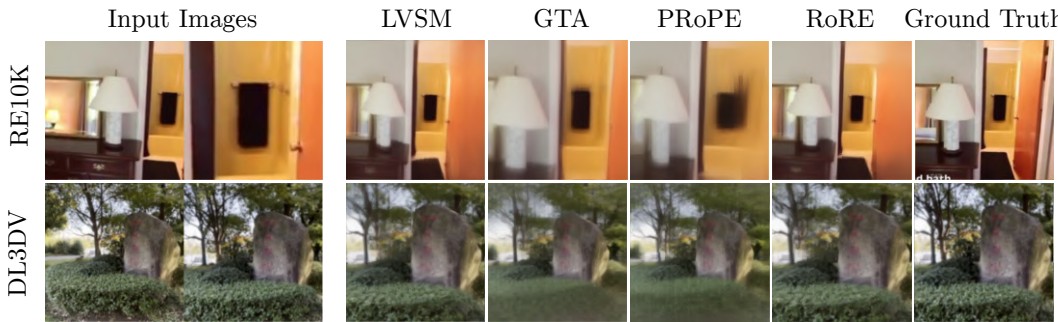

Figure 7: **Varying intrinsics in scene.** When the camera intrinsics varies within a scene without any additional training, we see both GTA and PRoPE fail to interpret the new cameras. The authors of PRoPE show that with training PRoPE is capable, however RoRE natively understands this.

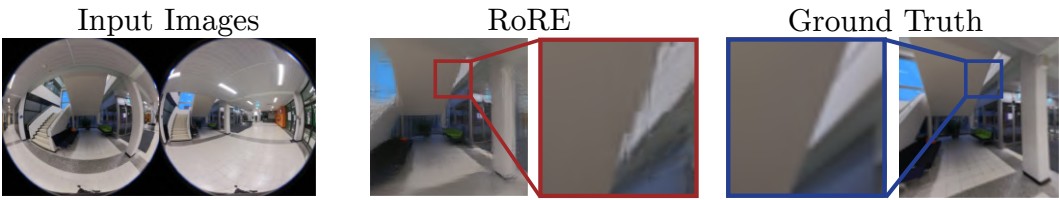

Figure 8: **Fisheye failure case.** Example failure case of RoRE of fisheye images, the local patches that RoRE produces can become misorientated. (red) shows an inset of RoRE's output, (blue) shows same inset from ground truth.

## A.2    ADDITIONAL RESULTS

### A.2.1    RORE RESULTS

Fig. 7 show qualitative results on the RealEstate10K and DL3DV datasets with varying intrinsics. Intrinsics were synthetically adjusted using cropping to increase the focal length. It can be observed that GTA and PRoPE fail to interpret the varying intrinsics values. LVSM performs better but looking at the RE10K example the rendering is partially warped. RoRE produces the best results.

### A.2.2    FISHEYE RESULTS

Fig. 8 shows a failure case of RoRE on fisheye imagery. This shows that while RoRE is able to render the global geometry quite well the local patch geometry is poorly reconstructed. One possible reason for this error is RoREs lack of ability to explicitly model a patches size and orientation. As these properties are implicitly estimated during training, when taken outside of that training domain, these estimations are likely less accurate.

### A.2.3    NETWORK SCALING

To confirm that our RoRE embedding is able to scale with larger networks, we doubled the network size from 6 to 12 layers and observed a substantial boost in performance, see Tab. 8 for numerical results. This level of increase in performance is typical for transformers (Jin et al., 2025), suggesting that our RoRE formulation doesn't negatively effect scaling performance.

### A.2.4    MULTI-MODAL RESULTS

**Masked Input.** The use of masked input tokens during training allows the model to handle partial occlusions and missing data at inference. By learning to reconstruct scenes from incomplete inputs, the network becomes robust to real-world scenarios where sensors may be partially obscured by dirt, water, or glare.

Table 8: **Scaling network size.** The RoRE embedding is able to scale to larger networks. We doubled the size of the network and observed a characteristic performance gain from transformer scaling.

| # Layers | # Parameters | PSNR(↑) | SSIM(↑) | LPIPS(↓) |
|---|---|---|---|---|
| 6 | 24M | 26.65 | 0.845 | 0.070 |
| 12 | 48M | 28.79 | 0.88 | 0.048 |

Table 9: **Progressive masking of input images.**

| Masked | RGB | | | Thermal | | | Depth | | |
|---|---|---|---|---|---|---|---|---|---|
| | PSNR(↑) | SSIM(↑) | LPIPS(↓) | PSNR(↑) | SSIM(↑) | LPIPS(↓) | AbsRel(↓) | RMSE(↓) | $\delta_1$(↑) |
| 0% | 21.494 | 0.634 | 0.254 | 20.481 | 0.807 | 0.172 | 0.060 | 0.025 | 0.964 |
| 10% | 20.875 | 0.618 | 0.269 | 20.021 | 0.799 | 0.181 | 0.063 | 0.025 | 0.962 |
| 30% | 19.612 | 0.576 | 0.303 | 18.933 | 0.779 | 0.205 | 0.067 | 0.027 | 0.957 |
| 50% | 18.287 | 0.525 | 0.350 | 17.755 | 0.752 | 0.239 | 0.076 | 0.031 | 0.945 |
| 70% | 16.913 | 0.477 | 0.409 | 16.367 | 0.722 | 0.286 | 0.105 | 0.042 | 0.884 |
| 90% | 14.902 | 0.448 | 0.496 | 14.183 | 0.693 | 0.366 | 0.207 | 0.082 | 0.537 |

We evaluate this robustness by progressively increasing the fraction of masked input tokens at inference, with quantitative results in Tab. 9 and qualitative examples in Fig. 9. Performance gradually declines as masking increases, but remains surprisingly stable up to moderate levels.

At 10–50% masking, reconstructions and depth predictions remain high-quality; even with 50% of input patches masked, key scene elements, such as the black box on the bottom shelf, are accurately recovered. A more noticeable drop occurs at 70% masking, where finer details are lost, yet the network still predicts plausible global structure and broadly consistent depth maps.

**Multi-Camera Reconstruction.** To evaluate the scalability and generalisability of our multi-modal transformer, we assess its performance with varying numbers and combinations of input images. Unlike previous models that assume a fixed number and modality, our framework can accept any number of views, mixing RGB and thermal inputs, constrained only by memory and compute at inference.

We perform two experiments, summarised in Tab. 10. In the first, we vary the number of inputs, using equal RGB and thermal splits (1 RGB + 1 thermal up to 4 RGB + 4 thermal). In the second, we fix six inputs and vary the RGB-to-thermal ratio to explore modality dominance effects.

Results show that increasing input number improves photometric and depth predictions, yielding more complete reconstructions with fewer artefacts. When varying modality composition, more thermal inputs enhance thermal fidelity but slightly reduce RGB quality, and vice versa. Depth estimates remain largely stable, performing slightly better when RGB images dominate, likely due to their higher spatial resolution and texture cues.

Table 10: **Varying number of input images and modalities.** Our proposed approach is able to handle a range of input images across a varying ratio of modalities.

| | RGB | | | Thermal | | | Depth | | | Encode Time | Decode Time |
|---|---|---|---|---|---|---|---|---|---|---|---|
| Input Images | PSNR(↑) | SSIM(↑) | LPIPS(↓) | PSNR(↑) | SSIM(↑) | LPIPS(↓) | AbsRel(↓) | RMSE(↓) | $\delta_1$(↑) | ms | ms |
| 1 (1-RGB) | 19.738 | 0.588 | 0.301 | - | - | - | 0.084 | 0.033 | 0.929 | 35.413 | 35.513 |
| 1 (1-thermal) | - | - | - | 18.610 | 0.783 | 0.215 | 0.100 | 0.036 | 0.909 | 34.312 | 34.902 |
| 2 (1-RGB, 1-thermal) | 20.470 | 0.605 | 0.280 | 20.068 | 0.803 | 0.181 | 0.062 | 0.026 | 0.958 | 37.493 | 40.662 |
| 4 (2-RGB, 2-thermal) | 21.751 | 0.638 | 0.246 | 21.008 | 0.816 | 0.161 | 0.053 | 0.022 | 0.966 | 47.489 | 44.182 |
| 8 (4-RGB, 4-thermal) | 22.767 | 0.656 | 0.223 | 22.104 | 0.833 | 0.140 | 0.044 | 0.019 | 0.979 | 95.566 | 54.249 |
| 6 (1-RGB, 5-thermal) | 20.927 | 0.615 | 0.269 | 22.082 | 0.833 | 0.143 | 0.051 | 0.022 | 0.971 | 71.211 | 49.486 |
| 6 (2-RGB, 4-thermal) | 22.013 | 0.642 | 0.241 | 21.829 | 0.829 | 0.146 | 0.050 | 0.021 | 0.972 | 71.451 | 49.569 |
| 6 (3-RGB, 3-thermal) | 22.409 | 0.649 | 0.231 | 21.580 | 0.825 | 0.151 | 0.049 | 0.021 | 0.975 | 70.531 | 49.441 |
| 6 (4-RGB, 2-thermal) | 22.730 | 0.656 | 0.223 | 21.209 | 0.819 | 0.156 | 0.048 | 0.021 | 0.975 | 70.524 | 49.403 |
| 6 (5-RGB, 1-thermal) | 22.916 | 0.661 | 0.218 | 20.534 | 0.808 | 0.168 | 0.049 | 0.021 | 0.975 | 71.594 | 49.488 |

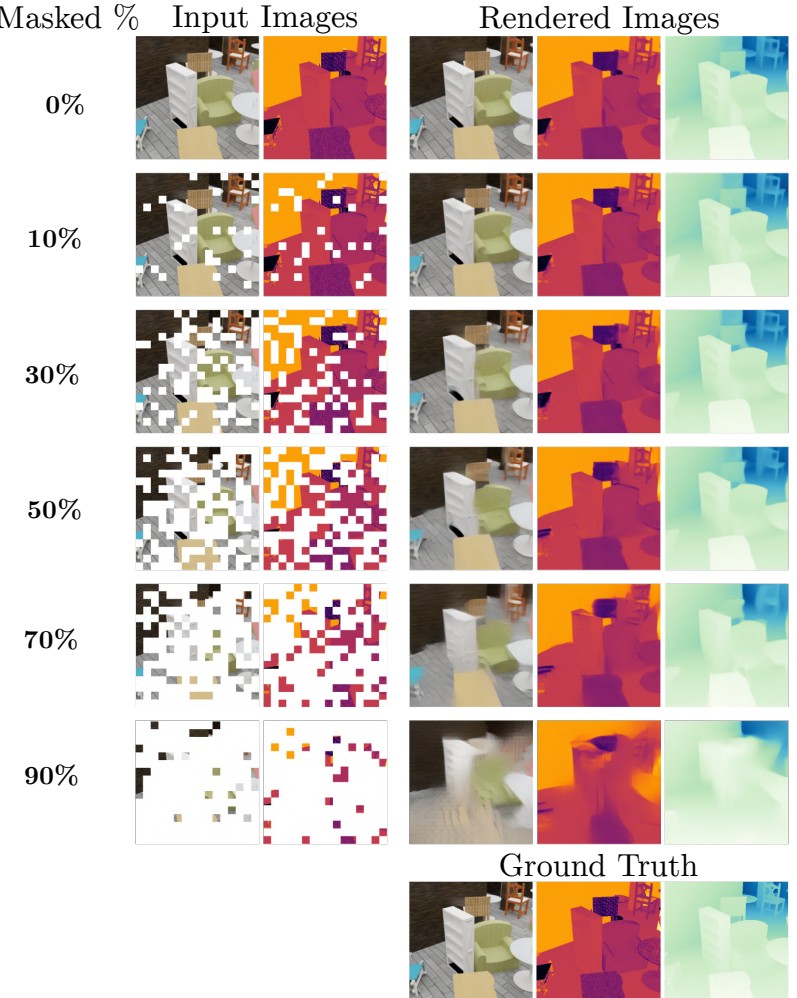

Figure 9: **Progressive masking of inputs.** Progressive masking of multi-modal inputs, the results show that the method is robust to the reduction in information.

## A.3 ENERGY USAGE

The development and training of large-scale machine learning models requires substantial computational resources, which in turn consume significant amounts of energy. In the interest of transparency and promoting sustainability in AI research, we tracked the total energy consumption associated with the experiments and model development presented in this work. Numerical values are provided in Tab. 11

Table 11: **Energy Consumption.** Energy consumption during model development and estimated equivalent emissions

| Metric | Estimate |
|---|---|
| Total Energy Used | 2554 kWh |
| $CO_2$-equivalent emissions | 1600 kg |
| Equivalent vehicle distance driven | 13,350 km |
| Equivalent household energy usage | 230 days |

Energy usage was monitored using the open-source `CodeCarbon` Python package Courty et al. (2024), which estimates energy consumption based on hardware usage and local electricity grid

intensity. Over the course of this work, the total energy consumed was approximately 2,900 kilowatt-hours (kWh), which is roughly equivalent to 230 days of typical household electricity usage.

By reporting these figures, we aim to provide context around the computational and energy costs of large-scale model development. We encourage the broader research community to adopt similar practices for monitoring and reporting energy consumption, supporting efforts toward more environmentally responsible and efficient AI research.

### A.4    GENERATIVE AI USAGE

ChatGPT was used as a proofreading tool to aid in grammer and sentence structuring. Where text was modified by generative AI, the content was reviewed for possible errors, inaccuracies, and bias. All ideas and content presented in this paper were conceptualised without the aid of AI.

