# OpenReview forum: "RoRE: Rotary Ray Embedding for Generalised Multi-Modal Scene Understanding"
_ICLR.cc/2026/Conference — ICLR 2026 Poster_

### Official Review · Reviewer_Q8Ad · 2025-10-16

**Soundness:** 3
**Presentation:** 3
**Contribution:** 3
**Rating:** 6
**Confidence:** 4

**Summary:**

This paper presents RoRE, a novel positional encoding technique for vision transformers. Built upon RoPE (rotatory encoding) that uses 2D patch index encoding, RoRE enhances each image patch using a Plucker ray with learned rotary positional encoding. The authors claim that doing so adds prior knowledge about the camera type into the model (regular cameras vs fisheye). Two innovations in RoRE are (1) learning the rotation frequencies of the positional embedding rather than fixed frequencies, and (2) applying asymmetric rotation to break symmetric biases in attention. The authors evaluated their model on five datasets, and RoRE shows to handle different camera/modalities with strong generalizability.

**Strengths:**

-	The method itself appears simple and thus should be easy to reproduce.
-	Learned 6-dimension positional encoding empirically aligns with the intuition that the frequencies should decay and shows more uniform patch-wise attention.
-	Experiments show performance gain compared to previous works (excepting concurrent work PRoRE).
-	The ablation studies cover most new components in the model.
-	The paper is well written and easy to follow. The differences from related works are explained clearly.

**Weaknesses:**

-	The idea of using Plucker rays as positional encoding has been introduced at least in prior work LVSM, which assigns pixel-wise Plucker ray embedding. (However, LVSM does not use positional encoding but simply passes the rays through a linear layer.)
-	A position value P_p is defined for each image patch. However, the pixels within an image patch could have different position values (rays). If just one value is used for an entire patch, then does the size of each patch matter?
-	In Table 4, the changes in metrics are fairly small. Recommend that authors further analyze why. The caption saying it is a boost is largely overstating.
-	In Table 5, the PSNR from thermal needs to be separated from the metrics for RGB.
-	The multi-modal scene understanding is a little bit distant from the main topic. It is hard to see what the authors want to show. Does the author want to demonstrate RoRE is a good sensor fusion tool? If so, this part needs much more detail and analysis. There isn’t any comparison to other works either.
-	The failure cases in Line 420 should still be shown in the paper.

**Questions:**

-	Line 161, recommend revising to: “, which has 3 position (or moment in the case of Plucker coordinates) dimensions t, and 3 direction dimensions d.” This improves readability.
-	In the qualitative results, what does each of the two input images represent? Recommend to explain somewhere.
-	Line 303, 381 has a grammatical mistake. “is can be see…”
-	Is the masking ratio a fixed value or does it follow a schedule during training?
-	In Figure 4, what are P1 and P2? Different modalities? Furthermore, what is being evaluated in multi-modal understanding? The predicted images seem to be from the same pose as the input, so are you testing image reconstruction?

---

> ### Author Response · Authors · 2025-11-22
> **Response to Reviewer  Q8Ad**
>
> We thank the reviewer for their valuable time providing both constructive feedback and the positive assessment. Below we address concerns and questions.
>
> ---
> **W1 - Plucker rays as an embedding representation**
>
> We agree that the idea behind using Plucker rays as the positional embedding representation is not new, many works use this representation. Our contribution is in expressing this Plucker representation in a relative embedding space. Relative embedding spaces have been demonstrated in this work and others to improve performance and generalisability.
>
> **W2 - Single ray for a whole patch**
>
> Yes, we only use a single ray to represent an entire patch, this is a limitation of this work, as it does limit the expressiveness of our method, this is also true for the orientation of the patch which is not expressed. In practice this orientation and patch size are likely implicitly estimated by the network. It would be some interesting future work to see if adding these extra parameters would improve performance. We have added a statement to the paper explaining this limitation as it is a relevant inclusion (Line 534).
>
> **W3 - Ablation study with marginal improvements**
>
> Although we initially expected the learned frequencies to yield stronger quantitative improvements, the learned variant converges to a frequency structure that closely resembles the handcrafted schedule. We hypothesise that this similarity explains the comparable performance between the two variants. Nonetheless, we believe that the learned formulation remains advantageous as it demonstrates that these rotations can be learned directly and offers greater generality without affecting performance or computation time.
>
> We choose the learned-frequency version as the final model because it removes hand-tuned hyperparameters and automatically adapts frequency allocation for position and direction components (Fig. 1). We have expanded the methodology section (Line 236) and the ablation discussion (Line 470) to better clarify this design choice, and have removed the overstated claims from Table 4.
>
>
> **W4 - Table 5, the PSNR from thermal needs to be separated from the metrics for RGB**
>
> We have formatted Table 5 (and others) to make the distinction between results clearer.
>
> **W5 - Multi-modal scene understanding distant from the main topic**
>
> The goal is to show a generalisable framework that can adapt to both multiple camera geometries and modalities. The main focus of this work is in the RoRE formulation so we acknowledge that the multimodal validation could be extended.
>
> **W6 - Fisheye failure cases**
>
> Good suggestion to include an example of the failures mentioned. We have included a figure in the appendix showing a close up of these fisheye artifacts (Line 825)
>
> ---
> **Q1 - Line 161 Revising**
>
>  Thanks for the recommendation, it has been revised.
>
> **Q2 - Explaining input images**
>
> We have added a statement (Line 314), explaining that the two input images shown in the qualitative results are two images of the same scene, from two different but known camera poses, that are used to generate renderings.
>
> **Q3 - Line 303, 381 Grammatical mistake**
>
> Thanks, fixed.
>
> **Q4 - Masking Ratio**
>
> The masking is a fixed value. However it would be interesting to see if a schedule could improve overall performance or convergence rate. The word ‘fixed’ has been included to remove ambiguity (Line 301).
>
> **Q5 - Figure 4 P1 and P2**
>
> P1 and P2 are images from two different poses 1 and 2. The objective is to show the model's ability to complete unobserved areas of one modality using information from other modality in a geometrically consistent way. These important clarifications of explaining symbols used and key takeaways have been added to figure caption (Line 454).
>
> ---
> We thank the reviewer Q8Ad again and welcome further discussion.

---

> > ### Comment · Reviewer_Q8Ad · 2025-11-25
> >
> > Thanks to the authors for the clarifications. My concerns are mostly addressed.

---

### Official Review · Reviewer_JXd5 · 2025-10-30

**Soundness:** 3
**Presentation:** 2
**Contribution:** 3
**Rating:** 6
**Confidence:** 3

**Summary:**

The paper proposes a novel positional encoding method for generalizable multiview vision transformers. The proposed Rotary Ray Embedding (RoRE) combines the benefits of relative positional encodings with ray-based position encoding methods, both of which have shown promising results in vision tasks and multi-view 3D scene perception. RoRE demonstrates intriguing robustness properties against changes in camera intrinsics and across heterogeneous sensor modalities. The experimental results validate the robustness and effectiveness of the proposed positional encoding method across various tasks.

**Strengths:**

- The method demonstrates robust performance on multiple tasks including novel view synthesis under varying focal lengths, barrel & fish-eye distortion, and multi-modal RGB-thermal fusion.
- The evaluation spans five datasets covering perspective imagery , fisheye cameras, and multi-modal setups, demonstrating breadth. The zero-shot generalization experiments (varying intrinsics, distorted inputs) are particularly valuable.

**Weaknesses:**

- **ablation study with marginal improvements**: Table 4 shows very small differences between ablation conditions (e.g., PSNR varies only 0.08-0.09 dB, SSIM differs by ~0.002-0.003). These marginal improvements raise questions about statistical significance of the reported differences and whether the proposed components genuinely contribute to performance.
- **Lack of interpretable analysis for learned frequencies**: Figure 1 shows position dimensions have larger rotation magnitudes than direction dimensions, but what does this mean conceptually? How does this relate to the semantic differences between position and direction?
- **Unclear practical motivation for RGB-thermal fusion**: While technically interesting, the paper doesn't establish compelling real-world applications for the joint RGB-thermal novel view synthesis. It would be beneficial to include real-world scenarios where joint RGB-thermal fusion would be valuable to help readers understand the practical significance of the method.

**Questions:**

- How is P = [t, d] computed for a patch? Is it the ray through the patch center? Average of all pixel rays?
- Why does standard RoPE produce biased attention that decays as position values diverge? How exactly does P = [t, -t+bias, d, -d+bias] "equalize the distance between two poses"?

---

> ### Author Response · Authors · 2025-11-22
> **Response to Reviewer JXd5**
>
> We thank reviewer JXd5 for the detailed feedback and for highlighting the strengths of our method. We address concerns and questions below.
>
> ---
> **W1 - Ablation study with marginal improvements**
>
> We agree that the numerical differences between the learned-frequency and fixed-frequency variants are small. To avoid ambiguity, we have explicitly included the no-RPE baseline (LVSM) in Table 4, which clarifies that both ray-based RPE variants provide gains over the no-RPE architecture.
>
> Although we initially expected the learned frequencies to improve quantitative performance, an interesting outcome is that they produce results nearly identical to the handcrafted version. We still believe this is notable, as it shows (i) that the rotational frequencies can in fact be learned, and (ii) that the learnt rotations converge to a meaningful multi-dimensional frequency structure (Fig. 1). The learned-frequency formulation also removes the need for manually selecting the hyperparameters used in handcrafted schedules, offering increased generality without negatively impacting performance or runtime.
>
> We choose the learned-frequency version as the final model for these reasons, and because it adapts frequency allocation differently for position versus direction (Fig. 1). We have expanded the methodology section (Line 236) and the ablation discussion (Line 470) to clarify this design decision.
>
> **W2 - Lack of interpretable analysis for learned frequencies**
>
> We have included a paragraph in the paper to provide additional intuition and analysis to help the reader interpret the results (Line 207) .
>
> Our intuition for the learned frequencies is in terms of the magnitude difference between the learnt rotation for position and directions; this is likely due to the distribution of direction values vs position values. Position values are normalised between zero and one for each example. The magnitude of the direction however for a given scene is less than one because of how the cameras move due to the requirement for overlap between frames . The more subtle differences in frequency decay are likely more complicated. Rotations are a much more complicated relationship between patches, having a faster decay show less importance on small frequencies, potentially indicating that more subtle changes are required for direction.
>
> **W3 - Unclear practical motivation for RGB-thermal fusion**
>
> We agree that the practical motivation should be made clearer and have added additional context to the introduction (Line 57).
>
> The main goal for this work is a general vision model that is general to camera geometries, modalities as well as number of images. We chose RGB-thermal joint modelling as it is relevant for several real-world applications, such as: infrastructure and asset inspection, where thermal cues reveal overheating or insulation failures, robotics and autonomy in low-visibility conditions (fog, smoke, darkness), search and rescue, where thermal imagery aids detection under occlusion.
>
> Beyond novel-view synthesis, the ability to represent RGB and thermal rays within a unified embedding space is also valuable for downstream tasks such as classification, anomaly detection, semantic segmentation, or multimodal scene understanding. These tasks often benefit from combining appearance cues (RGB) with temperature- or emissivity-based signals (thermal). By learning a shared geometric and multimodal embedding, RoRE provides a representation that can support such downstream applications without requiring separate architectures for each modality.
>
> ---
> **Q1 - How is the ray for a patch determined**
>
> Thank you for noting this missing detail of how the ray for a given patch is determined. This detail has been added to the paper (Line 170). We use the ray corresponding to the center of the patch, computed as the average of the rays of all pixels within the patch.
>
> **Q2 - RoPE Attention Decay**
>
> RoPE initially being applied to the task of NLP has by design been made to ‘focus’ its attention to tokens around the query, meaning tokens that are far away have been biased to have lower attention to the local meaning of words. We have included extra discussion on this point to the paper and have changed the terminology used from bias to shift, to improve clarity and understanding (Line 243).
>
> This is a natural property of the formulation [1](see Sect. 3.4.3, Fig. 2). In the context of 3D vision this property is undesirable. By providing a negative of positional values shifted by the maximum of the positional dimension, the overall magnitude of any positional value across the range of possible values remains consistent, resulting in no decay.
>
> We thank the reviewer again for the constructive feedback and are happy to address any additional concerns.
>
> ---
> [1] Su, Jianlin, et al. "Roformer: Enhanced transformer with rotary position embedding." Neurocomputing 568 (2024): 127063.
> Reviewer Q8Ad

---

> > ### Comment · Reviewer_JXd5 · 2025-11-28
> >
> > I appreciate the authors' feedback. My concerns and questions are mostly answered in the response.

---

### Official Review · Reviewer_VCbo · 2025-10-30

**Soundness:** 2
**Presentation:** 3
**Contribution:** 2
**Rating:** 6
**Confidence:** 3

**Summary:**

This paper proposes a Rotary Ray Embedding (RoRE) scheme that encodes camera parameters and images directly as ray-based scene tokens. RoRE feeds rays (camera origin + direction derived from intrinsics/extrinsics) into a transformer and applies a ray-conditioned variant of Rotary Position Embedding (RoPE) so that the model can handle perspective, fisheye, and RGB–thermal inputs with the same architecture.

**Strengths:**

The motivation and the solution—encoding everything into the ray—are clear to me. Most concurrent or prior works either stay with token-level ray maps but only for pinhole cameras, use relative camera encodings like PRoPE but are not multimodal, or handle thermal/RGB but not arbitrary cameras. This paper attempts to do all three at once to justify generalizability via ray-level embedding. Tables 2 and 3 support this by providing tasks under different camera intrinsics and different types of inputs (distorted or fisheye cameras). Table 9 indicates that multi-view rendering across modalities is possible.

**Weaknesses:**

“Improved generalization and cross-modal consistency” is not fully reflected across all experiments. I would consider raising the score if the authors can make some experimental designs more convincing.

* In Table 4, the PSNR result basically says: “if one keeps APE and adds our RPE, the numbers don’t really move.” That slightly weakens the claim that the ray component alone is doing the work. Are there more contrastive results on challenging tasks (e.g., a fisheye dataset)?
* Table 5 and Figure 5 are not convincing to me regarding why multi-modal training succeeds. The authors admit there are no known prior works (L364) for this particular problem. However, it might be possible to compare with a fusion-based transformer to assess whether transfer from a single modality to a different modality is feasible. Currently, the result has a PSNR around 22, which is not particularly promising.
* Again in Table 9, I wonder if “multiple images” implies rendering multiple views in one pass. If so, what is the difference between rendering 8 images once and rendering 1 image eight times, apart from time? Would rendering multiple RGB views confer any advantage if the backbone is drawn from RoRE/LVSM/GTA/PRoPE?

**Questions:**

* In L141–L142, is $\mathbf{R}_m^d$ a typo? Should it be $\mathbf{R}_m^n$?
* Any intuition on why initializing $\theta$ with a uniform distribution (L214) leads to an exponentially decaying final frequency in Figure 1?
* In Figure 6, should the RoPE module be replaced with the RoRE module? Can the experimental setup in Table 6—comparing different positional-embedding schemes—be understood as a replacement of this RoPE (or RoRE) module?
* A depth loss is added in L732–L750. How is ground-truth depth at pixel $i$ obtained? Is it required during training? What happens if $\lambda_{\text{depth}} = 0$?

---

> ### Author Response · Authors · 2025-11-22
> **Response to Reviewer VCbo**
>
> We thank the reviewer for their positive assessment and constructive feedback. We address the concerns raised below.
>
> ---
> **W1 - Table 4 Ablation Study**
>
> In Table 4, all variants do include our ray-based RPE instead differing how the rays are embedded. To clarify this, we have reworded the caption and include the no-RPE baseline (LVSM) in Table 4.
>
> While the results between the learned and handcrafted variations are very similar we still believe the learnt frequencies results are notable, as it shows (i) that the rotational frequencies can in fact be learned, and (ii) that the learnt rotations converge to a meaningful multi-dimensional frequency structure (Fig. 1). The learned-frequency formulation also removes the need for manually selecting the hyperparameters used in handcrafted schedules, offering increased generality without negatively impacting performance or runtime.
>
> We choose the learned-frequency version as the final model for these reasons, and because it adapts frequency allocation differently for position versus direction (Fig. 1). We have expanded the methodology section (Line 236) and the ablation discussion (Line 470) to clarify this design decision.
>
>
> **W2 - Confidence in Multi-Modal Results**
>
> Our approach relies on two key components to succeed at multi-camera, multi-modal reconstruction. First, the ray-based embedding used has been shown in our work as well as other to be robust across varying camera geometries. Second, we draw inspiration from prior works such as MultiMAE[1], which effectively embed multi-modal information in transformers via masked patch self-supervision. Together, these design choices enable a single model to reconstruct both geometric and modality-specific scene information. We have added a clarifying statement about the multi-modal methodology in the revised manuscript (Line 266).
>
> The goal of this evaluation is to demonstrate a generalisable framework that can adapt to both diverse camera geometries and modalities, which we believe we have shown. The MultiModalBlender dataset used in Table 5 presents more complex and dynamic motion than RealEstate10K, which explains the relatively lower metrics; we have added a note to clarify this (Line 509). The main focus of this work is in the RoRE formulation so we acknowledge that the multi-modal validation could be extended.
>
>
> **W3 - Table 9 Multiple Images**
>
> The “Multiple images” refers to multiple input views, not multiple output renderings. We have updated the caption and table heading text to improve clarity.
>
> Each forward pass renders one novel view. Increasing the number of inputs improves geometric coverage, which yields measurable gains in RGB, thermal, and depth reconstruction (Table 9). We include this result as our primary goal is a general method at work for a range of camera geometries, modalities as well as number of cameras (input images).
>
> ---
> **Q1 - Typo L141-L142**
>
> Yes, that should be $R^n_m$. We have corrected this.
>
> **Q2 - Initialising random rotations leads to exponentially decaying frequencies**
>
> This behaviour aligns with established research representing positions requires a spectrum of frequencies, with higher-frequency components capturing fine-grained variations and lower-frequency components capturing broader spatial trends.  The resulting learned decay therefore mirrors the intended multi-scale behaviour of classical positional embedding, providing evidence that the learnt embedding parameters can autonomously recover a meaningful and interpretable frequency structure. We added this intuition to the paper as it provides valuable context for why the results obtained are desirable (Line 207).
>
> **Q3 - Figure 6 RoPE**
>
> Yes, RoPE should be replaced with RoRE to show our method. We have updated the figure.  The baseline embedding methods replace that block, in the case of RoRE, PRoPE, GTA. In LSVMs case that block is removed instead as there is only absolute embedding.
>
> **Q4 - Depth Supervision**
>
> Ground truth depth is required during training, in order to get explicit depth output. We have added discussion to make clear the requirement for ground truth depth data in the main paper (Line 293).
>
> Our multimodal dataset has ground truth depth which was used. A good extension to this work would be to use self supervision to train this depth following work such as Zhou et. al [2]. Since we had depth we did not take the step to perform self supervision. If the depth loss $\lambda_{depth}$ is zero, the rgb and thermal would perform similarly, but the output depth will be arbitrary.
>
>
> Thanks to reviewer VCbo for the constructive feedback and are happy to address any further concerns they might have.
>
> ---
> [1] Bachmann, Roman, et al. "Multimae: Multi-modal multi-task masked autoencoders." European Conference on Computer Vision. 2022.
>
> [2] Zhou, Tinghui, et al. "Unsupervised learning of depth and ego-motion from video." Proceedings of the IEEE conference on computer vision and pattern recognition. 2017.

---

> > ### Comment · Reviewer_VCbo · 2025-11-26
> >
> > Thank author for their feedback. These changes address most of my concerns on clarification and motivation. I would remain my score. Two minor points:
> >
> > *    In Figure 6, there still exists one RoPE module, I think that is still a typo. Please fix it.
> > *    L550-551, "A link to an anonymous repository will be made available to reviewers in discussion forum." The reproducibility statement is not fulfilled yet.

---

> > > ### Author Response · Authors · 2025-11-28
> > >
> > > We are glad to have addressed some concerns and improved our paper.
> > >
> > > - Thanks for spotting that. It has been fixed.
> > > - Thanks for the reminder. A link has been provided above.

---

### Official Review · Reviewer_jq2U · 2025-11-01

**Soundness:** 3
**Presentation:** 3
**Contribution:** 3
**Rating:** 6
**Confidence:** 3

**Summary:**

This paper introduces a Rotary Ray Embedding approach that encodes image patches as rays to enhance multi-modal scene understanding. The proposed method demonstrates state-of-the-art performance and exhibits superior robustness under varying focal lengths and distorted input images

**Strengths:**

- The paper is well written and clearly organized.
- The proposed method achieves state-of-the-art performance and demonstrates superior robustness across challenging conditions.

**Weaknesses:**

1. Although the proposed method achieves state-of-the-art performance overall, it remains inferior to the concurrent work PRoPE in novel view synthesis results.
2. According to Table 4, all variants exhibit similar performance. In particular, the second variant and the full method achieve identical results across all three metrics, suggesting that the proposed learned frequencies may not contribute significantly.

**Questions:**

1. Although the proposed method achieves superior performance under varying focal lengths and distorted input settings, its performance on general novel view synthesis tasks is inferior to PRoPE. It is recommended to analyze the reasons behind this discrepancy.

---

> ### Author Response · Authors · 2025-11-22
> **Response to Reviewer jq2U**
>
> We thank reviewer jq2U for their time in providing thoughtful feedback and appreciate the positive assessment of our work. We address the concerns below.
>
> ---
>
> **W1, Q1 - Performance vs PRoPE**
>
> We appreciate the observation that PRoPE achieves slightly higher PSNR/SSIM on standard novel-view synthesis benchmarks. We also agree that a discussion around the difference between the methods is a valuable inclusion so have added an additional analysis to the results section (Line 430).
>
> The key difference is that RoRE is designed to provide a more general and geometry-agnostic positional encoding that remains robust across diverse camera families and modalities. This distinction explains the observed performance difference:
> - PRoPE uses a more constrained and camera-specific representation (projection-matrix-based relative encoding), which aligns closely with conventional perspective data and leads to strong in-domain performance.
> - RoRE embeds full rays with learned multi-dimensional frequency interactions, a strictly more expressive and geometry-agnostic representation. This flexibility yields markedly stronger generalisation under intrinsics changes, distortion, and fisheye inputs (Tabs. 2-3), but introduces a less constrained representational space, which can modestly reduce performance on narrow domains such as RE10K/DL3DV.
>
> We would also like to emphasise that PRoPE is concurrent work accepted by NeuRIPS during the review period.
>
>
> **W2 - Effect of Learnt Frequency**
>
> The learned frequency variant does exhibit very similar performance to the handcrafted frequency version. We acknowledge that we missed the opportunity to adequately explain our design choices. We have added clarification in the methodology (Line 236) and in the ablation discussion (Line 470) to better explain this design choice. We have also included an additional row to Table 4 showing results without any relative ray based embeddings.
>
> Although we initially expected learned frequencies to improve quantitative results, an interesting outcome is that they instead produce performance nearly identical to the handcrafted version. We still believe however that it is a notable result because:
> - it shows that these rotations can be learnt, and
> - that the model successfully learns a meaningful frequency structure (Fig. 1).
>
> We also believe that the benefits outlined in the methodologies are still valid, particularly removing the need for manually selecting hyperparameters of the handcrafted frequencies. This motivated us to select the learned-frequency formulation as the final version of RoRE due to its increased generality without negatively impacting performance or processing time.
>
> We thank the reviewer again for the constructive comments and are happy to address any remaining concerns.

---

> > ### Comment · Reviewer_jq2U · 2025-11-24
> >
> > Thank the author for their feedback. My concerns are addressed and I decide to keep my rating unchanged.

---

### Author Response · Authors · 2025-11-22
**General Response**

We sincerely thank all reviewers for their time and thoughtful feedback, which has helped us improve our paper. We are encouraged by the generally positive reception of our work and provide detailed responses to each reviewer below.

A revised version of the paper, with the current changes highlighted in blue, has also been uploaded. We plan to further refine the paper based on ongoing feedback.

---

> ### Author Response · Authors · 2025-11-28
>
> An anonymized repository for our RoRE method is provided [here](https://anonymous.4open.science/r/RoRE-896C).

---

### Meta-Review · Area_Chair_sP3E · 2026-01-06

**Summary:**

All reviewers agree that the presented submission is solid, giving it a rating of 6/6/6/6 = 6 average (weak accept).
The rebuttal has addressed some concerns, without changing the scores significantly. Importantly, no reviewer stated new concerns, i.e. the original scores stand.

Following the reviewer's assessment, I recommend acceptance.

**Reviewer Concerns:**

Ssee above.

**Reviewer Scores:**

Reviewer jq2U: 6 unchanged (explicitly kept the score post-rebuttal)
Reviewer VCbo: 6 unchanged (explicitly kept the score post-rebuttal)
Reviewer JXd5: 6 unchanged (concerns were "mostly answered")
Reviewer Q8Ad: 6 unchanged (concerns were "mostly mostly addressed")

---

### Decision · Program_Chairs · 2026-01-26

Accept (Poster)